# The Rise of Vietnamese Nuns: Views from the Buddhist Revival Movement (1931–1945)

Ninh Thị Sinh

Department of History, Hanoi Pedagogical University 2, Phuc Yen 15900, Vietnam; ninhthisinh@hpu2.edu.vn

**Abstract:** In this article, with the aim of better understanding the development of Vietnamese Buddhist nuns, the period of the Buddhist revival movement is investigated. This event is considered a turning point for Vietnamese Buddhism. In addition, it will help to shed light on the status of Vietnamese nuns. In this article—which is mainly based on archival documents kept in the National Overseas Archives (the French colonial archives held at the Archives Nationales d'Outre-Mer) and the National Archives Center I, Buddhism periodicals, and memoirs—the status of Vietnamese women during the French colonial period is clarified, as well as the positive effects of the colonial regime in regard to the change in women's perceptions. Then, the differences in the nuns' situation in three regions are analyzed. Finally, an exploration is conducted into the rise of nuns during the revival movement and the emergence of reformist nuns. Indeed, it is reformist nuns that have shaped the image of modern Vietnamese nuns. Moreover, they also created a direction by which the following generations could continue along, as well as playing an important role in the Vietnam Buddhist Sangha.

**Keywords:** Buddhist revival movement; Buddhist nuns; monastic status; Vietnam

## 1. Introduction

Unlike its neighboring nations in Southeast Asia (which follow Theravāda Buddhism and there is no presence of Buddhist nuns) there are, in fact, nuns in Vietnamese Buddhism. According to the Most Venerable Thich Tri Quang (Acting Supreme Patriarch of the Patronage Council of the Vietnamese Buddhist Sangha (*Quyền Pháp chủ Giáo hội Phật giáo Việt Nam*)), in 2008, the number of Vietnamese nuns was not only larger than that of monks, but many of them were highly educated and increasingly active in various fields[1]. Some famous nuns of the twenty-first century include: Bhikkhunī Tri Hai (1938–2003), renowned in the field of education; Bhikkhunī Nhu Duc (Vien Chieu Zen Monastery, Long Thanh), recognized for her monastery management; and Bhikkhunī Hue Giac (Quan Am Monastery, Bien Hoa), a pioneer in the protection of the ecological environment, and who planted and cared for over 400 hectares of forest. Vietnamese Buddhism includes both Mahāyāna (*Bắc tông* or *Đại thừa* in Vietnamese) and Theravāda (*Nam tông* or *Tiểu thừa*) branches (including Khmer Theravāda and Kinh Theravāda) in its development. Furthermore, from the mid-twentieth century onward, a local branch, known as the Mendicant sect (*hệ phái Khất sĩ*), was also developed. According to the Mahāyāna and Mendicant sects, to become a nun one needs to go through the following stages: the probationary period (two years, the time when a woman takes refuge in the Three Jewels and follows the five precepts); after two years, she accepts ten precepts and becomes a Sāmaṇeri; for the next two years, she studies six rules and becomes a Sikkhamānā. Finally, after these two years, she accepts 348 precepts, thereby becoming a Bhikkhunī. One has to be at least a Sāmaṇeri to be classified as a disciple of the Buddha. By Khmer Theravāda tradition, it has only been monks who could lead religious activities and have their contributions fully recognized. Additionally, the Kinh Theravāda has now ordained women, but they accept just eight precepts, and there is still no full ordination for them. Therefore, in the current

Vietnamese context, nuns are monastic women who are ordained and have received ten or more precepts, according to Mahāyāna and Mendicant traditions[2].

The year 2008 was a significant year for Vietnamese Buddhist nuns, as this is when they established their own organization: the division in charge of nuns (*Phân Ban Ni giới*) under the Central Committee of the Vietnam Buddhist Sangha (*Ban Tăng sự Trung ương Giáo hội Phật giáo Việt Nam*). It is an organization with a clear structure and operating rules that unifies the leadership and management of nuns throughout the country[3]. However, the division is not the first organization of Vietnamese Buddhist nuns. In 1956, nuns had already established an organization in southern Vietnam called the South Vietnam Bhikkhunī Sangha (*Ni bộ Nam Việt* in Vietnamese). It must be stated that this was a notable event in the history of Mahāyāna Buddhism in Vietnam. Despite this, northern Vietnam is considered to be the cradle of Vietnamese nuns who follow Mahāyāna Buddhism. This rather confusing fact leads to the question: Why? In order to address this question, we must investigate the period of Buddhist revival.

Although several studies have essentially unified viewpoints regarding the beginning of the Vietnamese revival movement—which emerged in the 1920s (Woodside 1976, pp. 192–200; Marr 1981, p. 304; Nguyen 2007; DeVido 2007, pp. 250–69; Ngo 2015)—this is not the case regarding the end of the movement. The Buddhist revival did not really "end" in any of the years 1945, 1951, 1954, 1963, or 1975, which marked key turning points for Vietnam and Vietnamese Buddhism (DeVido 2007, p. 251). However, within the scope of this article, we have only investigated the period from 1931 to 1945 for the following reasons: Firstly, the study of the revival movement is achieved mainly through Buddhology/Buddhist associations, namely, the Cochinchina Buddhist Studies Association (1931, Cochinchina), the Luong Xuyen Buddhist Studies Association (1934, Cochinchina), the Annam Buddhist Studies Association (1932, Annam), and the Tonkin Buddhist Association (1934, Tonkin). These associations were founded in the 1930s, with the first association being the Cochinchina Buddhist Studies Association, which was established in 1931. With the founding of this association, the movement gained legal status (Nguyễn 2008, p. 615). By 1945, when Vietnam gained independence, these associations were either defunct, or had changed their names, as well as their statutes. For example, the Annam Buddhist Association changed its name to the Vietnam Buddhist Association (*Hội Việt Nam Phật học*), and the Tonkin Buddhist Association also changed to the Vietnam Buddhist Association (*Hội Việt Nam Phật giáo)*. Secondly, we surveyed the Buddhist periodicals *Từ Bi Âm* (Sound of Compassion), *Viên Âm* (Sound of Perfection), *Duy Tâm Phật học* (Mind-Only Buddhist Studies), and *Đuốc Tuệ* (Torch of Wisdom), which served as the official organ for the abovementioned associations, as well as the majority of suspended editions in 1945[4].

This article utilizes both Buddhist periodicals—which were established during the revival movement—and archival sources. Their purpose was to shed light on the differences in the nuns' situation between the three regions during the colonial period, as well as to shed light on their voices within the context of the revival movement. Some examples of Buddhist periodicals are *Pháp Âm* (Voice Dharma, 1929), *Phật Hóa Tân Thanh Niên* (1929), *Từ Bi Âm* (1932–1945), *Viên Âm* (1933–1945, 1949–1953), *Duy Tâm Phật Học* (1935–1943), *Đuốc Tuệ* (1935–1945), *Tam Bảo* (The Three Jewels, 1936–1938, Annam), and *Tiếng chuông sớm* (Sound of Early Bell, 1935–1936). By surveying this collection of periodicals, we found 56 articles written by nuns. Regarding archival materials, we consulted the documents at the National Overseas Archives (Aix-en-Provence, France) on the collection of GGI (Gouvernement général d'Indochine) and RSTNF (Résident Supérieur du Tonkin Nouveau Fonds). At the National Archives Center I, Hanoi, we consulted the collection of GGI (Gouvernement général d'Indochine), RST (Résident Supérieur du Tonkin), and MHN (Mairie de Hanoï). At the National Archives Center II (Ho Chi Minh City), we investigated the collection of Goucoch (Gouvernement de Cochinchine). Lastly, at the National Archives Center IV (Da Lat), we focused on the collection of the Central Vietnam governors. Thanks to these archives, we were able to find records of Buddhist associations, pagodas, and surveys on the activities of Buddhist associations. During the survey, we found two

dossiers: 2405 (collection of RSTNF, Archives Nationales d'Outre-Mer, France) and 3722 (collection of MHN, National Archives Center I, Hanoi, Vietnam), which concern nuns in Tonkin. Among them, dossier 2405 is a very important and meaningful record (Archives Nationales d'Outre-Mer 1943). Indeed, it mentions *Notices individuelles sur les chefs de pagodes* (individual instructions on the chiefs of pagodas) secretly made in 1943 in the provinces of Tonkin by the French authorities. Each instruction has the following basic information: name; age; address; educational and Buddhist background; membership in the Tonkin Buddhist Association; relationship with officials and relatives; religion; material status; the importance of the monastery; its wealth; the degree of influence of pagodas; and the number of members, i.e., from the abbots to the followers. Underneath each instruction is the signature and seal of the village mayor *lý trưởng*, or district chief *tri huyện*, or French resident *công sứ*. The instruction below [Figure 1] allows us to visualize the specific information that was outlined in the file:

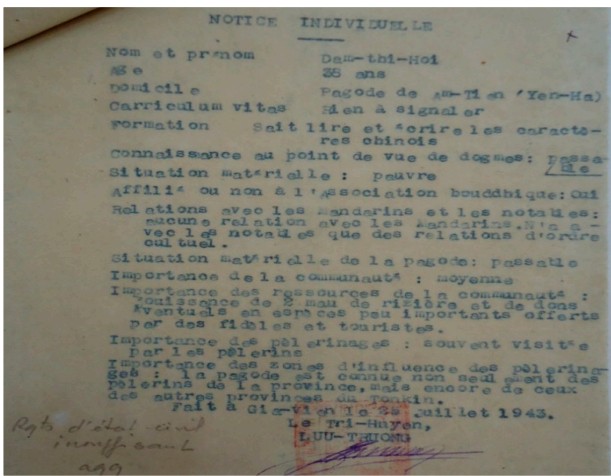

**Figure 1.** The instruction concerning individual nun Dam Thi Hoi (dossier 2405).

The information regarding the monks and nuns in the provinces, however, was not complete in these archives. For example, the Hai Duong province in *bordereau d'envoie (schedule of sending)*, dated 10 August 1943, mentioned seven individuals, but, in fact, there are no specific instructions pertaining to seven individuals to be found in the dossier. However, by handling the instructions concerning individuals in the archives, we found important information regarding the Tonkin nuns. In addition, we also effectively utilized the memoirs of the nuns at that time, such as the memoir of the nun Dieu Tinh.

This article focuses on nuns that practiced in the Mahāyāna tradition. In the history of Vietnamese Buddhism, Mahāyāna was the earliest (during the second or the third centuries CE) that appeared. The Kinh Theravāda and Mendicants were, in contrast, established relatively late. Kinh Theravāda was founded in Vietnam in 1938 by Hộ Tông (in the name of Lê Văn Giảng), and the Mendicant sect was founded in 1944 in the south by the great patriarch Minh Đăng Quang. We first present Vietnamese women during the French colonial period. This period was chosen as it helps to emphasize new factors in cultural and social life that were brought upon by the colonial regime, and which affected the perception of Vietnamese women. Then, an analysis of the differences in the situations of nuns in the three regions during the French colonial period is performed. Finally, we discuss the rise of reformist nuns within the Buddhist revival movement, during the period 1931–1945.

## 2. Vietnamese Women's Situation during the French Colonial Period (1884–1945)

Prior to the French colonial invasion, Vietnam was a united kingdom with an absolute monarchy, under the rule of the Nguyen emperors. With this ruling system, the emperor was considered the "son of the sky" (*Thiên tử*), who ruled the people on behalf of the heavenly god; he was also the representative of tradition and the spirit of the nation.

The Nguyen dynasty used Confucianism as its ideological foundation, thereby affording Confucianism a profound influence on all aspects of social life. Under the influence of Confucianism, the status of women was considered insignificant (Đặng 2008, p. 28). Indeed, they suffered many injustices due to the concept of "valuing men above women" (*trọng nam khinh nữ*). Dieu Khong, a woman from the Royal family, was reported to have said that "since the [idea of] Confucianism was enthusiastically adopted by people all over our country, women had to restrict themselves to the regulations of the Confucian doctrine, so a woman's duty was only in the family, and [she] would not be expected to know or participate in anything other than that" (Diệu 1935c, p. 40). In her memoirs, the nun Dieu Tinh also mentioned the low status of women on several occasions: "In social life, women sometimes suffer more . . . Becoming a wife, her husband only considers her a housekeeper, a reproductive machine, or a recreational item. Is it not insufferable." (Diệu 1926, p. 5)

From the middle of the nineteenth century, the Dai Nam kingdom of the Nguyen emperors gradually came under the rule of the French colonial empire. After the French attack on the port of Da Nang in 1858, Saigon was occupied in 1859. The South then became a French colony under the name of Cochinchina in 1862. The rest of the territory was divided into Tonkin and Annam under the protectionist regime in 1884 (Brocheux and Hémery 2001, pp. 33–52). The defeat of the Hue dynasty by the French army started the decline of Confucianism and represented the starting point for the entry of Western cultural and ideological factors into Vietnam. These political shifts brought numerous changes in all aspects of the economic and cultural lives of different social classes in Vietnam, including among women.

With the birth of the Franco-Vietnamese education system, education was no longer the sole prerogative of men. The year 1907 marked the first establishment of girl schools. It was a breakthrough, not only in terms of education, but also especially in terms of ideology, which was positively received by the Confucianist Sinic class (who had previously considered educating girls as useless) (Nguyễn 2020, p. 58). At the end of World War I in 1918, schools for girls were available in all three regions of Vietnam. The number of female students continuously increased (Trịnh 2019, pp. 163, 171, 182). From a social perspective, it must be said that female education contributed to changing women's status in society (Nguyễn 2022, p. 36).

The *Quốc Ngữ* (Romanized Vietnamese) script also gained popularity during this period (Brocheux and Hémery 2001, pp. 221–25). Indeed, this was one of the "most fundamental legacies" of colonial education. When it was first created in the seventeenth century, this script did not initially extend beyond the scope of evangelism. It was primarily used for educational and teaching purposes during the colonial period that enabled the *Quốc Ngữ* script to gain widespread acceptance with the masses.

The birth and later popularity of the press (especially the Vietnamese-language press) affected Vietnamese perceptions. The first newspaper in *Quốc Ngữ*, called the Gia Dinh newspaper (*Gia Định Báo*), was first established in 1865 in Saigon by the French colonial authority in Cochinchina. Moreover, it was Pétrus Ky, the famous scholar, who then continued the publication. Over time, the number of newspapers in *Quốc Ngữ* increased. During the 1920s and 1930s, the Vietnamese printing press achieved great growth in both the number of its publications and the variety of its content (McHale 2004, pp. 18–19). In particular, other than in the general press, there appeared specialized newspapers for women. In the developmental period of 1930–1935, in all three regions, women's newspapers appeared. Among them, the most famous is the *Phu Nu Tan Van* (News of Women) newspaper. As a weekly newspaper, which existed from 1929 to 1939, it mainly focused on "the masses, dealing with everyday issues" (Huỳnh 2016, p. 223). *Phu Nu Tan Van* was "appreciated by all circles, not only widely popularized in Cochinchina but also received by readers in Tonkin and Annam with much good affection" (Huỳnh 2016, p. 220). It can be said that, thanks to the press, women had access to diverse information, ranging from politics, economy, culture, science, religion, and women's issues.

A significant change in Vietnamese society during the French colonial period was represented by the birth of modern cities and the formation of Western urban lifestyles. Under the influence of French colonial exploitation, urbanization rapidly increased in Vietnam. Hanoi and Hai Phong became first-class cities, similar to Saigon (Brocheux and Hémery 2001, pp. 179–81). These cities quickly became political, economic, and cultural centers with buildings and offices of the colonial government, factories, business properties, shops, and trading units. In the provinces, class-2 and class-3 urban areas and towns appeared. Moreover, Western-style urban life led to new ideas (Thanh and Chân 1997, p. 16).

These factors created great changes in the socio-cultural life of women. In addition to the majority of women living in rural areas—who were still influenced by Confucian thought—a section of women (especially women in urban areas) were able to go to school and access diverse sources of information pertaining to the news. This new situation changed perceptions and thinking.

## 3. The Situation of Vietnamese Nuns during the French Colonial Period (1884–1945)

Although Vietnam has a long tradition of bhikkhunīs dating from the twelfth century (DeVido 2007, p. 279), there are significant historical differences between regions. The nuns in the north have a longer history than those in the central and south regions (Nguyễn 2012, p. 813). In the north, not only do nuns have a long tradition, but they also have their own temple. We obtained two archival dossiers and two books on which to base this statement. Dossier 3722—in the collection of MHN, National Archives Center I (Hanoi, Vietnam)—provided information on the status of the Quan Su Temple in 1933[5]. The dossier outlined that, before the temple was ceded to the laypeople, this temple was managed by two nuns, Nguyen Thi Doan and Nguyen Thi Tan (National Archives Center I 1933). This fact was also mentioned by the monk Tri Hai, who initiated the Buddhist revival movement in Tonkin in his *Memoirs of the founding of the Vietnamese Buddhist Association* (*Hồi ký thành lập Hội Phật giáo Việt Nam*) as follows: " . . . occasionally monk Thai Hoa takes me to visit Quan Su Temple, Hanoi. The abbess there at that time was nun Nguyen Thi Doan" (Trí 2016, p. 28). Another archival document, dossier 2405, provided more information about the Tonkin nuns. First, the number of nuns who were abbess of their temple were detailed [Table 1]. The rest of the results are summarized in the following table.

**Table 1.** The number of nuns who were abbess of their temple in Tonkin.

| Province | Number of Abbesses (Out of Total) |
|---|---|
| Hưng Yên | 11/34 |
| Hà Nam | 2/14 |
| Thái Bình | 40/144 |
| Ninh Bình | 18/70 |
| Bắc Ninh | 2/30 |
| Vĩnh Yên | 4/8 |
| Phúc Yên | 0/7 |
| Phú Thọ | 13/26 |
| Quảng Yên | 2/16 |
| Kiến An | 0/6 |
| Thái Nguyên | 0/5 |
| Tuyên Quang | 0 |
| **TOTAL** | **92/360** |

Through the process of researching the instructions of the abbesses in this dossier, we also determined that, other than ordinations of monks, some monastics possessed master nuns [Figure 2]. Below is an example to support this:

**Figure 2.** Nun Dang Thi My was a disciple of nun Tue (dossier 2405).

Nun Dang Thi My (born in Tho Khoi, Bac Ninh province) was abbess of Vy Thanh Temple (canton of Quat Luu, Binh Xuyen district, Vinh Yen), and nun Le Thi Lien was the abbess of Thuy Van Temple (Hac Tri district, Phu Tho). There was also a disciple of the master nun, Tue, at the Son Thi Temple (Lam Thao district, Phu Tho province).

From the information provided in the dossier, we determined a basis for the conclusion that, in the Tonkin region during the French colonial period, nunneries (*sơn môn ni*) existed. This observation is also reinforced by the information provided by Nguyen Lang (known by the religious name Thích Nhất Hạnh) in *Việt Nam Phật giáo sử luận* (History of Buddhism in Vietnam): "In Hanoi, there is a large nunnery in Hang Than street that has been established for many generations; called nunnery Am (*sơn môn Am*) . . . In Ha Dong, in Khoang village, there is another large nunnery called nunnery Khoang (*sơn môn Khoang*). This is also a large patriarchal nunnery (*tổ đình*)" (Nguyễn 2012, p. 813). In addition, Nguyen Lang also noted that these nunneries are training places for nuns. At nunnery Am: "Every year the nuns gather here to study Buddhism in summer, sometimes more than a hundred nuns" (Nguyễn 2012, p. 813). Furthermore, the large nunneries mentioned above, dossier 2405 also demonstrated that some local temples served as training places for monks and nuns. The Phu Vien village Temple (Gia Thuy, Gia Lam, Bac Ninh) "was a school for the training of monks and nuns"[6], or, regarding the Dong Nhan Temple (Dai Trang village, Van Mau, Vo Giang, Bac Ninh), it "was a place of training for monks and nuns little frequented"[7].

During such a long tradition, there were eminent nuns in Tonkin. Nguyen Lang noted that the Venerable Dam Soan (?–1968) was the first nun to be invited to the Hue royal palace in order to teach Dharma to empresses and concubines (Nguyễn 2012, p. 813). Moreover, dossier 2405 tells us about four other nuns, including nun Dam Yen (Bac Ninh), Bui Thi Doai (Thien Phuc Temple, Truc Phe village, Thuong Nong canton, Tam Nong, Phutho), a nun (unnamed) who was the abbess of Chung Quang Temple, and the nun Le Thi Lien (Thuy Van village, Hac Tri, Phu Tho), who all possessed an in-depth understanding of Buddhism.

However, through careful examination of the 92 instructions concerning nuns, we found that—at the Buddhist level—the vast majority were of elementary educational level, and there was even a sizable proportion of nuns who were labeled as being "illiterate" and "knowing the prayers by heart". Therefore, it is not surprising that nun Tâm Nguyệt wrote in *Đuốc Tuệ* in 1937 that nuns could not explain the basic concepts of Buddhism: " . . . If someone asks what the Buddhadharma (*Phật pháp*) is, and what Buddhist practice (*tu hành*) is, only a few people can answer correctly, some even say the Buddhadharma is the Buddhadharma itself, and Buddhist practice means repairing the garden to plant onions

for sale to earn some money" (Tâm 1937, p. 3) ( . . . *"có ai hỏi thế nào là Phật pháp, là tu hành, thật ít có người trả lời đúng được, thậm chí có vị trả lời Phật pháp là Phật pháp chứ là gì, còn tu hành là cốt sửa sang vườn giồng hành cho tốt để bán lấy tiền tiêu"*).

Regarding the education level of the nuns, the results of the survey carried out by the colonial authority in 1943 showed that: 67/92 knew Chinese characters; 5/92 knew *Quốc Ngữ* and Chinese characters; 18/92 were illiterate; and 2/92 were without information.

Generally, a nun had a narrow influence and was confined to a village, mostly having an "insignificant influence in the community" or "non-influence on the believers". The fact that some nuns did not specify their names (along with their influence on the community and followers), leads to strong evidence that the Tonkin nuns lived a quiet life within that period.

There are very few documents concerning nuns in Annam and Cochinchina during, as well as before, the French colonial period. As such, there is very little information regarding nuns in these two regions. In Annam, the book *Lược sử ni giới Bắc tông Việt Nam* (A Brief History of Vietnamese Mahāyāna Buddhist Nuns) stated that: "In the late nineteenth century, there was no nun temple and [its] own system in Hue, the Nguyen kings established temples in the palace, called king temples, so that women who work in the Royal Household practiced Buddhism, sometimes listening to the sutras. The nuns who have ordained and studied are all dependent on the monks." (Như 2009) The document also records that nun Dien Truong (1863–1925) was the first bhikkhunī of Annam. She came into the temple in 1898 with the Venerable Hai Thuong Cuong Ky in Tu Hieu Temple. In 1910, she was ordained as a bhikkhunī at the Quang Nam transmission ceremony. After the building of Truc Lam Temple, the nun Dien Truong invited the Venerable Giac Tien to be the Abbott of the temple. In addition, she "set up a private nunnery at Truc Lam temple and gathered a number of other nuns to study, such as nun Chon Huong, Dieu Huong, and Giac Hue" (Thích and Hà 2001, p. 436). In 1928, the first nunnery appeared in Hue; i.e., Dieu Vien Temple. This was a temple founded by Ms. Ung Dinh and the nun Dieu Khong (who was not yet ordained at that time). Later, Dieu Huong was invited to be the chairperson.

Particularly in Cochinchina, it is difficult to identify the bhikkhunī who initiated the nun tradition. In the works on the history of Vietnamese Buddhism by Van Thanh, Nguyen Tai Thu did not mention the Vietnamese nuns (Vân 1974; Nguyễn 1988). As an aside, it can be considered that the *Việt Nam Phật giáo sử luận* (Vietnamese Buddhist History) of Nguyen Lang is the first work that mentions the nuns in Cochinchina. In his work, he introduced three Cochinchina nuns, including Dieu Tinh (1910–1942) (who is an important character that we will describe in greater detail in the next section), Chi Kien (1913–2007), and Dieu Ninh (1914–?). The common factor between them all is that they were all born in the second decades of the twentieth century and were ordained in the 1930s. The book *Lược sử ni giới Bắc tông Việt Nam* (A Brief History of Vietnamese Mahāyāna Buddhist Nuns) also reported—in addition to the nuns Dieu Tinh and Dieu Ninh—the nuns Dieu Tan (1910–1947), Hong Nga-Dieu Ngoc (1885–1952), and Chon Ngan-Nhu Hoa (1910–1989) (Như 2009). Another common factor between the nuns mentioned in the two works is that they all left home to study with the monks. Nuns Hong Nga-Dieu Ngoc, Hong Khoai-Huu Chi, Hong Tho-Dieu Tinh, Hong Tich-Dieu Kim, Buu Thanh, and Hong Lau-Dieu Tan were all disciples of the Venerable Nhu Hien Chi Thien, also known as the Venerable Phi Lai (Như 2009, pp. 44–45). The nuns Chon Ngan-Nhu Hoa, Chon Niem-Nhu Ngoc, and Chon Vinh-Phuoc Hien were disciples of the Venerable Giac Ngo Chanh Qua (Kim Hue temple, Sa Dec) (Như 2009, p. 64), who was an active monk in the Buddhist revival movement. Although there is no archival material for this, there is the work of Tran Hong Lien, as well as articles in the Buddhist periodicals at that time—which both provide information on the situation of the nuns in many aspects. Tran Hong Lien, in her book *Đạo Phật trong cộng đồng người Việt ở Nam Bộ-Việt Nam từ thế kỷ XVII đến 1975* (Buddhism in the Vietnamese community in Southern Vietnam from the seventeenth century to 1975), wrote: "The number of nuns appeared more and more" (Trần 1995, p. 94). Nun Dieu Tinh, a nun

at the time, further mentioned "a lot of women ordained" (Thích 1935, p. 19). Although the nuns in Cochinchina were "more and more numerous", the reality was that they did not have their own temple, and thus, must have followed the monks. This was reflected by nun Dieu Tinh and Mrs. Tran Nguyen (Louise) in articles published in *Viên Âm* and *Từ Bi Âm* at the time. In *Viên Âm* No. 17, 1935, nun Dieu Tinh wrote: " . . . women ordained a lot, but there was no temple to live in and nuns had to take refuge in monks' temples" (Thích 1935, p. 19). Supporting the statement by Dieu Tinh, Mrs. Tran Nguyen not only detailed the actual situation, but also commented on the consequences of the "living together of monks and nuns", "without any strict separation between monks and nuns, so there are many tragedies of nuns, which serves as a mouthpiece for the worldly people to slander Buddhadharma" (Trần 1936, p. 36). Perhaps such a sentiment was also the reason why nun Huệ Tâm chose to end her life when she was young, even though she had demonstrated full enthusiasm and dedication to the Dharma[8].

In terms of education, the first nun class in Cochinchina was organized by nun Hong Nga-Dieu Ngoc at Giac Hoa Temple in Bac Lieu in 1927. However, this class was only maintained for one year. When reading Dieu Tinh's memoirs, we can imagine the difficulties facing nuns in learning the Buddha's teachings at that time.

Nun Dieu Tinh, whose secular name was Pham Dai Tho, was born in 1910 in a well-to-do Catholic family with seven siblings in Yen Luong Dong village, Hoa Dong Ha canton, Go Cong. When possessed with the desire to live a meaningful life and be helpful to other people, she decided to become a nun. She committed to monasticism with a monk at Tan Lam Temple (Tan Son Nhat, Gia Dinh) at the age of 14, after trying to convince her parents. She thought she could study in peace, but she did not expect to face so "many obstacles". After the daughter of Mrs. *Hội đồng* had fallen in love with her (because the daughter thought Dieu Tinh was a man), a young modern man had also flirted with her. It was unfortunate that she did not meet a good teacher and friend, as her actual teacher "held the broomstick and hit on Dieu Tinh's head" due to the fact that she was boiling water while talking to the other novice. In another instance, the teacher "burned all the sutras I [Dieu Tinh] wrote for a long time and kicked me out immediately" because she prepared the offerings until 2 am and then fell asleep due to being too tired. These matters made it such that the young novice "feared the teacher like a tiger" and "was sad about learning, not knowing whom to learn from" (Diệu 1926).

For the nuns, the monastic life mainly involved "doing good work for the pagoda" (Thích 1933, p. 20). Another source, dating from 1938, recorded that nuns were engaged with "coining and serving tea for the monks, and the nuns also took on very heavy and very hard responsibilities: cooking rice, washing dishes, sweeping the temple, sewing robes for the religious men. In a year with twelve months, in a month with thirty days, in a day with twelve hours, there was not any free time to enter the temple to admire the Buddha's face, not to mention time for study or research . . . " (Như 1938, p. 27). (*sự cạo gió hầu trà cho ông sư, lại còn lãnh lấy cái trách nhiệm rất nặng nề, rất cực nhọc: nào nấu cơm đun nước, nào rửa chén quét chùa, nào khâu quần vá áo cho các ông chúng đạo, một năm mười hai tháng, một tháng ba mươi ngày, một ngày mười hai giờ, tưởng không có chút nào hở tay rảnh việc mà vào chùa cho thấy mặt Phật, có đâu nói đến sự học hỏi nghiên cứu*). Therefore, when compared to what would be their work at home, the duties of nuns did not differ much from the responsibilities of women in society. They were bound to the work of the house, kitchen, and garden. The nuns' monastic life was simply the epitome of women's lives in society. Consequently, the nuns had "the name and image of the monastic people only" (Tâm 1937, p. 3).

With the honor of being the cradle of Vietnamese nuns, nuns in Tonkin—despite their cloistered life and their level of preliminary Buddhism—clearly had more favorable conditions than the nuns in Annam and Cochinchina. The Tonkin nuns had their own temple. Moreover, even large nunneries had nuns who could understand Buddhist dharma, as well as studied Buddhism during the Three-Month Summer Rains Retreat Courses. Due to this, the history of the nuns in the Annam and Cochinchina regions began later. There was no specific temple for nuns. They had to follow male masters and stayed in the monks'

temples. There was no school for nuns to study Buddhism. The key point is that the nuns of the three regions existed separately and had no relationship with each other, except for a few individuals who moved between regions to study Dharma. In addition, their responsibilities were associated with the kitchen and garden.

## 4. Buddhist Revival Movement—The Opportunity for Buddhist Nuns to Rise Up

The Buddhist revival movement was an international movement initiated in India (Nguyễn 2012, p. 531), which then spread to other Asian countries, including Vietnam. Directly influenced by the Chinese Buddhist revival movement, especially from Master Taixu[9], it was initiated in Cochinchina (South Vietnam) by Khanh Hoa and his monastic friends (Nguyen 2007, p. 120), and started in the 1920s. The Buddhist revival movement went through two main phases: the campaigning phase, which featured in the *Quốc Ngữ* newspapers during the 1920s; and the period of revival practice, associated with the Buddhist associations throughout the three regions of Vietnam in the years 1931–1945. The Buddhist revival movement appeared in the context of Vietnamese society during a time when there were also many Western cultural elements coming into competition with traditional ones. It was also during the era of the erosion of Confucianism, the mainstream ideology of the Kings of the Nguyen dynasty, the decline of Buddhism, and the penetration of Western democratic ideas due to the reform activities of progressive patriotic intellectuals, such as: Phan Chau Trinh; Huynh Thuc Khang; Tran Quy Cap; Luong Van Can; the development of modern writing and printing techniques; and the widespread promotion of *Quốc Ngữ*. Therefore, the movement reflected not only a religious character, but also a social character. The different specific action plans of the various associations were aimed at some of the following goals, as explained by Nguyen The Anh: "In general, the leaders of the movement firstly set themselves the goal of interpretation, explanation of the teachings and scriptures in *Quốc Ngữ*, because before there had only been a small number of monks who could understand the Hán (Chinese) Buddhist scriptures, which led to a hindrance of the widespread dissemination of Buddhist teaching. On the other hand, with belief, Buddhism can only be revived when the tranquility is regained, therefore, they demanded the purification of the monastery, establishment of monastic disciplines, and development of modern schools to train a new generation of educated and virtuous monks, capable of taking a spiritual leadership role in a volatile society. Finally, they wished to give Buddhism a systemized organization, allowing this religion to stand firmly to its rival, Christianity, which was regarded as an instrument of colonial authority" (Nguyễn 2008, p. 615). To achieve such goals, the Buddhist Associations published Buddhist periodicals in *Quốc Ngữ*, translated sutras and Buddhist books from Hán to *Quốc Ngữ*, opened Buddhist schools to train talents, taught Dharma, built temples, and organized social charity activities. With these activities, the revival movement provided new ways and opportunities for nuns to raise their voices and express their ideas.

The nuns were no longer quiet, but active. They appeared in Buddhist media where they participated by writing articles and gave Buddhist sermons in public. This was an unprecedented phenomenon in the history of Vietnamese Buddhism. Their doctrinal activities were recorded in the Buddhist journals of the time (including *Từ Bi Âm, Viên Âm, Đuốc Tuệ, Duy Tâm*), as well as the nuns' authorship in the three regions. Nun Dieu Tinh, for example, gave four lectures. The first time was on the occasion of the Congress of Cochinchina Buddhist Studies Association (1935). The second time was on the occasion of the inauguration of Hai An Temple on 30 August 1936. The third time was at Thien Phuoc Temple (Soc Trang), the head office of the Tuong Te Buddhist Association, on the occasion of its inauguration (13 July 1936). The fourth time was at Quoc Cong temple in Hung Yen on the occasion of her trip to the north in 1938. Furthermore, Nun Hue Tam lectured at Dong Quang Temple in 1935 (Tonkin); nun Dieu Vien gave a lecture on the occasion of the inauguration of the Annam Buddhist Studies Association in Da Nang (1936); and nun Tam Nguyet gave a lecture on the anniversary of Buddha's birthday in Quy Nhon (1937).

The frequency of media appearances by nuns is shown in the statistics [Table 2], obtained through articles in the Buddhist periodicals at that time:

**Table 2.** A list of articles by nuns in the main Buddhism periodicals.

| Periodical | Total Number of Articles by Nuns | Authors | Time Appearance |
|---|---|---|---|
| *Từ Bi Âm* (1932–1945, Cochinchina) | 20 | Diệu Tịnh, Huệ Tâm, Diệu Tâm, Diệu Ngôn, Diệu Minh, Như Ý, Diệu Nhựt | 1933–1940 |
| *Duy Tâm Phật học* (1935–1943, Cochinchina) | 4 | Diệu Hữu, Diệu Hường, Diệu Tịnh, Diệu Tánh | 1937–1939 |
| *Viên Âm* (1933–1945, 1949–2953, Annam) | 27 | Diệu Viên, Huệ Tâm, Diệu Không, Diệu Tịnh, Diệu Phước, Diệu Tu, Tâm Nguyệt, Diệu Hồng, Diệu Hòa, Tâm Diệu | 1934–1938 |
| *Đuốc Tuệ* (1935–1945, Tonkin) | 5 | Tâm Nguyệt, Diệu Tịnh, Đàm Như, Đàm Hữu, Đàm Hướng | 1937, 1938, 1944 |

The statistical table shows that the Cochinchina nuns were at the forefront of "showing up" in Buddhist media. The person who started it was nun Dieu Tinh with the article "Complaint of a Buddhist Nun" (*Lời than phiền của một cô vãi*) in *Từ Bi Âm*, No. 27, published on 1 February 1933. The nuns in Annam were next (1934), and the last were the nuns in Tonkin (1937). Nuns from Cochinchina and Annam appear more frequently in the media than those from Tonkin with regard to both the date and the number of articles. The nuns in Cochinchina continuously featured for 9 years in two main journals, and the nuns in Annam continuously featured for 8 years; whereas the nuns of Tonkin only appeared in a scattered manner for 3 years. In this respect, nun Dieu Tinh is the most active. Her name appeared in all four periodicals, from Cochinchina to Tonkin.

Nuns were not only appearing more frequently in the Buddhist press, but they also expressed interest in many issues of Buddhism, from fundamental Dharma-based issues to topical revival issues. In particular, they paid great attention to issues directly related to the nuns at that time. These facts proved that there were nuns who could be qualified, enthusiastic, and responsible. Responsible for their own monastic life, for the Dharma, for the revival career, and for nuns in general. A brief overview of some of the major issues that were of concern to the nuns at that time clearly demonstrates this.

The articles "We should believe in Buddhism, Buddhism is not superstitious" (*Chúng ta nên tín ngưỡng Phật pháp, tín ngưỡng Phật pháp không phải là mê tín*) (Huệ 1935b, pp. 17–31); "Advising people to learn Buddhism" (poetry) (*Khuyên người học đạo*) (Diệu 1935a, pp. 54–55); "How to learn Buddhism" (*Thế nào là học Phật*) (Diệu 1935b, pp. 43–46); and "Buddhism is not in conflict with the present situation" (*Phật giáo không mâu thuẫn đối với cục diện ngày nay* (Diệu 1935d, pp. 12–15) were all articles that had the characteristics of defending Buddhism and referring to the benefits of learning Buddhism. Furthermore, nun Dieu Khong explains the purpose of studying Buddhism: " . . . studying Buddhism is to know one's own mind, to live a perfect life, to work for life, to be useful for life . . . do not have to look for the peaceful and quiet realms to hide" (Diệu 1935b, p. 40). Hue Tam, in the article "We should believe in Buddhism, Buddhism is not superstitious", criticized two misconceptions about Buddhism. The first misconception was the notion that Buddhism is an "ambiguous" religion, whereas the second considers that "Buddhism is a religion that relies on the divines, so it is only worship, so that whenever everyone has difficulties in life or property, then pray for the accident to pass" (Huệ 1935b, p. 21). In order to solve these problems, nun Hue Tam emphasized the good spiritual "values" of Buddhism. Buddhism covers the meaning of "very broad equality", and "Buddhism does not force us to believe without thinking". Buddha taught people to "focus on the mind" (*chú trọng cái tâm*). From the abovementioned Buddhist values, she came to the general conclusion that Buddhism

is not only dedicated to "monks, nuns" and "believers", but it is the fortune of the whole of humanity.

With the articles "The Buddhist Associations should be united" (*Các hội Phật học nên hiệp nhất*) (Huệ 1935a, pp. 4–11), "A few words" (*Đôi lời thỏ thẻ*) (Diệu 1936e, pp. 61–63), and "Buddhism today must have a radical reform" (*Phật học ngày nay phải có sự cải cách triệt để*) (Diệu 1939, p. 527), the nuns responded to the problems of the revival movement. Nun Hue Tam proposed that, in order to render the revival effective, "Buddhist associations should unite" into a "big organization" (*đại tùng lâm*), which meant creating a common institution for Buddhism in the country. Furthermore, this agency would then carry out revival activities in four aspects: distinguishing Buddhism from non-Buddhists; correcting the Sangha; regulating the obligations of the laypeople; and controlling Buddhist propaganda agencies. Supporting this concept, nun Dieu Tanh, in the *Duy Tâm*, also advised on the issue of establishing a "General Buddhist Association" (*Phật giáo Tổng hội*) (Diệu 1939, p. 531). This was a significant and bold idea that referred to a major issue within the revival movement. However, this idea was not feasible due to the fact that Vietnam was divided into three regions (*ba kỳ*). Meanwhile, nun Dieu Tu gently offered her opinion regarding the way Buddhist periodicals worked at that time, and how they should instead be employed. With respect to the Buddhist journals, instead of using the press to "vilify", "reproach", and "reprimand" each other, they should instead affect the image of Buddhism, they should "use gentle words to promote each other", and they should reflect the actions of "forgery monks" (*tà sư*) in order to protect the Dharma.

Another issue of concern to the nuns was found in women's issues concerning Buddhism. Nun Dieu Vien pointed out the "hesitancy" of women regarding Buddhism and explored what could be the likely causes (Diệu 1936f, pp. 11–19). As a result, she introduced "Benefits of Buddhist studies for women" (Diệu 1934, pp. 11–15). Meanwhile, nun Dieu Minh introduced various methods of propagating the Buddhadharma to women of different ages. At a young age, Dharma preachers could use poems and rhyming songs about Buddhism, they could draw Buddhist pictures, and they could collect and translate famous quotes about Buddhist ethics in order to instruct children on how to learn by heart. When they are old, the preachers could guide them to recite the Buddha's name, preach the sutras, the laws of cause and effect, and perform charity work (Diệu 1936a, 1936b, 1936c). Finally, nun Dieu Phuoc advised on the meaning of Buddhist ethical practice for the moral perfection of women in the article "Women and Buddhism" (Diệu 1935e, pp. 23–25): " . . . absorbing the Buddhist morality, then in the family, sisters are good wives, good daughter-in-law, and mothers who have all the virtues; In society, women are selfless who benefit the nation and the people, and strive for a good evolution for society" (Diệu 1935e, p. 23). Along these lines, nun Dieu Phuoc promoted the image of a modern woman who absorbed Western ideas, but still retained the traditional values of Vietnamese women. This is different from the "half-civilized" (*văn minh nửa mùa*) women, imitating the new trend from the West without thinking.

In the Buddhist periodicals, nuns were mostly concerned with nuns' issues. There are eleven articles related to this issue that cover the following aspects: opening a school for nuns, establishing a nunnery, building a nun community, and publishing a Buddhist newspaper dedicated to nuns. The issue of nuns' education was the central issue, thereby attracting the attention of nuns Dieu Tinh[10], Nhu Y (Như 1938, pp. 26–30), Dieu Huong (Diệu 1938a, 1938b), Tam Nguyet (Tâm 1937), Dam Nhu (Đàm 1944c), Dam Huu (Đàm 1944b), and Dam Huong (Đàm 1944a). On discussing this issue, first of all, nun Dieu Tinh emphasized the role of learning in order to enhance education level, and emphasized that, in Dharma practice, for nuns: "Learning is like a torch, only understanding Dharma you know the way to monastic practices" (Thích 1933, p. 19) (*Sự học thức là gậy là đuốc, có thông hiểu mới biết lối tu hành"*). Learning and understanding the teachings of Buddha could help them to "propagate the Dharma and do good for being" (*hoằng pháp lợi sinh*) (Diệu 1935f, p. 42) or "help the Sangha to revive the Buddhism" (*giúp ích cho phái tăng già lo việc trùng hưng Phật pháp)* (Thích 1933, p. 22); essentially, that a monastic life was not a wasted life. In

order to achieve this goal, they expressed their wish that the Buddhist Associations open Buddhist schools for nuns. In Cochinchina, nun Dieu Tinh, in as early as 1933, proposed her aspiration that: "Each Venerable monk should focus on the education of our nun sisters to quickly achieve wisdom and virtue" (*mỗi vị cao tăng đại đức nên chú trọng về sự giáo hóa cho chị em chúng tôi mau thành tài đạt đức*) (Thích 1933, p. 22). In Tonkin, in 1937, nun Tam Nguyet (just a Sāmaṇeri at the time) desired to study well. Therefore, she "does not hesitate to write this article, hoping that people [in the Tonkin Buddhist Association] with the heart to think about Buddhist Dharma, Sangha, raise your hands to bring knowledge to our sisters, and please do not be biased, despising our nun sisters" (Tâm 1937, p. 4).

It should be further explained that the "school for nuns", as mentioned by nun Tam Nguyet, was not a traditional school, but a Three-Month Summer Rains Retreat Course[11]. She, instead, referred to opening a modern school for nuns similar to the monk schools. The modern school for monks was opened by the Tonkin Buddhist Association in 1936, with four levels: primary, secondary, university, and college. Monks studied all year with four semesters, including both Buddhist Sutra and non-Buddhist scriptures[12]. In addition, the nuns also encouraged each other on how to "arrange to have time to study" (Đàm 1944c, p. 5).

Whether due to the fervent wishes of the nuns or not, in Cochinchina, there were many monks, such as Giac Ngo Chanh Qua and the Venerable Khanh Hoa (a pioneer of the movement to revive Buddhism in Vietnam), who opened classes for nuns, or allowed nuns to attend certain classes. For example, there was: the Kim Huê Buddhist School for monks and nuns; Phước Huệ home Buddhist school for nuns (opened by Zen Master Giac Ngo Chanh Qua); and the Vĩnh Bửu Buddhist Class for nuns that was opened by Zen Master Khanh Hoa (Như 2009, p. 68). Not only did these institutions rely on venerable monks, but the nuns in Cochinchina also took the initiative in opening the schools. We would like to emphasize the role of nun Dieu Tinh in opening schools for nuns in Cochinchina. Her contribution of teaching and opening schools for nuns can be seen in the statistics below [Table 3][13]:

**Table 3.** A list of Buddhist schools attended by nuns.

| Year | Names of School/Class | Location |
|------|----------------------|----------|
| 1933 | Giác Hoàng Buddhist Summer Rains Retreat Course for both monks and nuns. This was the first Buddhist Summer Rains Retreat Course to allow nuns to enroll. Nun Diệu Tịnh was given the post of lecturer. | Giác Hoàng Pagoda (Bà Điểm) |
| 1934 | Home Buddhist Class for Nuns (*Lớp gia giáo* in Vietnamese). Here, nuns studied the Buddhist scriptures in Chinese and basic Buddhist studies. | Hải Ấn Buddhist nuns's Temple (Gia Định) |
| 1934 | *A Three-Month Summer Rains Retreat Course.* Nuns Diệu Tịnh and Như Thanh were teachers. This was the first Buddhist nun school solely organized by nuns. | Thiên Bửu Temple (Lái Thiêu) |
| 1940 | Tân Hòa Home Buddhist School for Nuns, three months. The nuns Diệu Tịnh and Diệu Không were the teachers. | Giác Linh Temple (Tân Hòa, Sa Đéc) |
| 1941 | Linh Phước Home Buddhist School for Nuns opened by nun Diệu Tịnh after having been invited by Mrs. Bang Biện. | Linh Phước Temple (Cai Khoa, Sa Đéc) |

In Tonkin in 1938, save for the Three-Month Summer Rains Retreat Courses, a secondary and primary class for nuns was opened at But Thap Temple (Ninh 2020, p. 207). In Annam, although there were no articles on *Viên Âm* about opening schools for nuns, there were, in fact, schools for nuns that were also established within the framework of a Buddhism revival led by the Annam Buddhist Association. In 1932, the first nun school opened at Tu Dam temple. In 1934, the Dieu Duc nunnery was established as a training and education institution for nuns in Annam (Như 2009, p. 25). Regarding nuns' activities (such as opening schools and building nun temples in Annam) during this time period, it is impossible to overlook the role of the eminent nuns in Hue, especially the Venerable Dieu

Khong (1905–1997), who was of Royal origin. She was adept at Confucianism and Western studies. In addition, she made great contributions to nuns in Annam and helped to build nunneries; she also founded many schools for nuns[14].

According to instances in the Buddhist press, if the nuns in Tonkin were only interested in opening new schools, nuns in Cochinchina were also interested in issues that were both urgent for and oriented toward nuns, such as the construction of a nunnery. This is understandable due to the fact that the nuns in Cochinchina did not have their own temples. Nun Dieu Tinh was a person that had great success in mobilizing the monks to build Từ Hóa nunnery. When discussing the campaign to build Từ Hóa, in the article *Một bức thư dài xin hỏi ý kiến chị em nữ lưu* [*A long letter asking for female sisters' opinions*] published in *Viên Âm*, she wrote: "In the month of April 1935, we personally came together with monk Thích Từ Phong and monk Thích Pháp Ấn, to request the Venerable Master Trang Quản Hưng to apply for the establishment of a nunnery, under the name Từ Hóa Tự, in the village of Tân Sơn Nhì, Dương Hòa Thượng district, in Gia Định . . . This temple was built with the money of the local people and all nuns. The temple, although established, has not yet satisfied our hopes" (Thích 1935, p. 22). In 1935, Dieu Tinh moved the temple to Tân Sơn Nhất and changed its name to Hải Ấn Ni tự. The temple consisted of three compartments and two wings (which were built of brick and roofed with tiles). Furthermore, it was inaugurated on 30 August 1936, the fifteenth day of the seventh lunar month (Diệu 1936d, p. 44). Although somewhat simple, constructing the temple demonstrated the nun's hearty efforts for their community. It was from this first nunnery that the nuns had their own base from which to practice and study. In addition, it helped overcome the situation of "monks and nuns living together". As such, from this foundation, the nuns continued to build more nunneries.

The issue of building an organization for nuns was also raised. Nun Dieu Tinh was the first and most frequent person to draw attention to this issue. In 1935, on the occasion of the Congress of the Cochinchina Buddhist Studies Association, nun Dieu Tinh called for nuns to "adjust the nuns to be orderly, build a union, then must gather all material resources and human resources to take care of the restoration of the Buddha Dharma . . . " (Diệu 1935f, p. 42). Furthermore, the same year, in *Viên Âm*, in an article addressed to the nuns, she continued to mention the issue of the nuns being able to contact each other in order to form a community of their own: "I wish that the nun sisters studying Buddhism would completely abandon their divisive nature, stop identifying different masters or sects, setting a border between pagodas and temples, which have been a regrettable practice of the elder monks for a long time. From now on, each of the three regions and each province has its own temple for nuns. Moreover, I hope all nuns would learn to love each other, unite their different ideas, follow the same set of rules, and organize a Bhikkhunī Sangha [in which we might] study the scriptures together, strictly adhere to the precepts to maintain the Buddha Dharma in this period of weak Dharma and strong superstition, and thus fulfill our aspirations" (Thích 1935, p. 22). Then, in *Từ Bi Âm*, No. 148, in 1938, she continued to call for "establishing an association of nuns" (Diệu 1938c, pp. 29–33). In *Duy Tâm*, in 1938, Dieu Huong also emphasized the solidarity of the nuns, citing that "the sisters should join the union, separate between nuns and monks, arrange orderly to avoid the harm of disintegration and division" (Diệu 1938a, p. 274).

Cochinchina nuns were pioneers in expressing the voice of Vietnamese nuns in the revival movement. In the press, they mentioned the manifold problems of the nuns, not only in providing a reaction to the conditions of nuns in Cochinchina (which were somewhat unfavorable compared to the Tonkin and the Annam nuns), but also in demonstrating how the nuns in that area were influenced by an open and modern lifestyle under the influence of Western civilization. In the history of Vietnam, Cochinchina is a land where Confucians did not have as profound an influence as in Annam and Tonkin. However, Cochinchina experienced the effect of Western civilization earlier than those in Annam and Tonkin because Cochinchina became a French colony first. Considering the nuns at that time, nun Dieu Tinh was the most eminent. This was not only because of her specific actions, but also because of the issues she initiated that were related to the nuns. Dieu

Tinh was a rare case among the Vietnamese nuns in the 1930s and 1940s of the twentieth century. According to her memoirs, she was born in a Catholic family in Cochinchina, but soon received a French–Vietnamese education where she was literate in both *Quốc Ngữ* and French; as a result, she was familiar with Western culture (Diệu 1926). It was one of the more favorable starting points for her when she committed herself to Buddhism. She possessed a profound awareness of the path of study, as well as the status of nuns. Furthermore, it was the "obstacles" in her monastic life that focused her direct attention on the study of nuns, the construction of nunneries, and the promotion of the issue of solidarity among nuns. Her short life was devoted to her wish to "elevate the status of nuns to the level of monks" (Nguyễn 2012, p. 779). Therefore, she became an example to others, and was often mentioned in the Buddhist press whenever the monks promoted the spiritual practice of the nuns. *Từ Bi Âm* itself also presented a poem to the nuns praising Dieu Tinh, as follows:

> Speech acts must be supreme
> So that in a female body
> Hundred lifetimes forged a sword of wisdom
> One hand opened the door to a life of dust and heat
> Leading three to five groups of religious mates
> Defeating six or seven parts of forgery monks . . . (Từ 1935, p. 45)[15]

### 5. Conclusions

The period of French colonial rule (1884–1945), in addition to its negative consequences[16], also brought about positive effects. In particular, these positive effects ranged from the cultural to the ideological, as well as more broadly in terms of knowledge. These factors affected all social classes, including women. The opening of girl schools and opportunities to be exposed to Western European ideological movements changed aspects of social awareness, as well as women's perceptions of their own roles and statuses, thereby raising feminist-based issues. It must be stressed that nuns are a community of Vietnamese women. In a society influenced by Confucianism, their lives at home were associated with the "three followings and four virtues". When they left home, they were influenced by the concept of "regarding monks higher than nuns" (*quý tăng tiện ni*) in the temple. Although the specific situation was not the same in the three regions, in general, the nuns had low status, blurred images, lived quiet and closed lives, and were confined within the monasteries, as well as the villages.

The revival movement emerged, and with it, nuns boldly stepped out from the traditional old and narrow social framework to become reformist nuns. They appeared in the Buddhist media. They preached in public at major ceremonies, such as the congress of the Buddhist Association, the inauguration of temples, the establishment of the Buddhist Association, or on the anniversary of the Buddha's birthday ceremony. Through articles and lectures, they demonstrated a high level of Buddhist studies, reflected on the situation of the nuns, and expressed their wish to have schools for nuns, as well as to have their own temples. They called on nuns to unite in order to build their own union. The key issues of concern were urgent issues that needed to be resolved, but they were also issues that orientated on the development of nuns in the following periods. When comparing the nuns in the three regions, the voices of the nuns in Cochinchina were more vibrant. Among them, the most important character was the nun Dieu Tinh, a young, enthusiastic nun with an established and well-educated background in Buddhism. She was a pioneer who raised her voice, making the case for the rights of nuns. Their achievements created not only a foundation for nuns to continue to rise up in the following periods, but also contributed to solving feminist issues in Vietnamese society in the early decades of the twentieth century.

**Funding:** This research received no external funding.

**Institutional Review Board Statement:** Not applicable.

**Informed Consent Statement:** Not applicable.

**Acknowledgments:** I wish to thank Nicola Schneider and Ester Bianchi for inviting me to participate to the conference "Gender Asymmetry in the Different Buddhist Traditions Through the Prism of Nuns' Ordination and Education" (Perugia, 16–17 May 2022). I am grateful to Nicola Schneider, and Trent Walker for their willingness to read an earlier draft of this article. My gratitude goes to the anonymous reviewers for their suggestions and corrections. I would also like to thank the Hanoi Pedagogical University 2 for supporting me to carry out this research under grant number HPU2.2022-UT-06.

**Conflicts of Interest:** The author declares no conflict of interest.

## Notes

1.  https://thuvienhoasen.org/p80a4699/2/vai-tro-cua-ni-gioi-viet-nam-trong-xa-hoi-hien-nay-thich-tri-quang accessed on 22 November 2022.
2.  https://phatgiao.org.vn/ni-gioi-ho-la-ai-d43957.html (accessed on 22 November 2022).
3.  http://vinhnghiem.de/news/index.php?nv=news&op=Dieu-le/Noi-qui-Phan-ban-Dac-trach-Ni-gioi-Trung-uong-231 accessed on 7 November 2022.
4.  Từ Bi Âm (1932–1945); Viên Âm (1933–1945, 1949–1953); Đuốc Tuệ (1935–1945); Duy Tâm (1935–1943).
5.  Head office of the Tonkin Buddhist Association, now it is the head office of Vietnamese Buddhist Sangha Central.
6.  *La pagode est une école pour la formation des bonzes et bonzesses* in French (Archives Nationales d'Outre-Mer 1943).
7.  *La pagoda est un lieu de formation des bonzes et bonzesses peu frequentée* in French (Archives Nationales d'Outre-Mer 1943).
8.  This Buddhist nun was from Tonkin (Northern Vietnam) but in order to learn the Dharma she had to travel to Cochinchina (Sourthern Vietnam) to study with other monks, such as Zen Master Khanh Hoa and Zen Master Huệ Quang. Compared to other Northern Buddhist nuns at that time, she was one of the few who had an academic background before ordination. Coming from a "reputable and wealthy" family, she was educated in *Quốc Ngữ* and French, and spent two years living in China with her family. Because of her good academic background and other personal qualifications, Huệ Tâm progressed very quickly on the path of Buddhist cultivation. Although the number of the articles she produced was not terribly significant, the content clearly shows her high level of knowledge on Buddhism, as well as the awareness of a young nun towards the problem of the revival movement. However, it remains unclear why she chose to end her life by drowning herself at Ngao Châu beach. It is tragic indeed about Huê Tâm. See Ninh (2019, pp. 89–100).
9.  Taixu visited Vietnam two times in 1928 and 1940. In addition, his ideas, and the activities of the Chinese Buddhist reform movement were already well-known in Vietnam via Taixu's writings and his disciples' propagation. For more information on the influence of Chinese Master Taixu on Buddhism in Vietnam See (DeVido 2009, pp. 413–58; Nguyen 2007, pp. 127–28).
10. Lời than phiền của một cô vãi (Complaint of a Buddhist Nun). *Từ Bi Âm* 27: 18–23; Nên tổ chức trường Phật học để giáo dục phụ nữ không? (Should a Buddhist school be organized to educate women?). *Từ Bi Âm* 148: 29–33. This article was republished in *Duy Tam* 32: 355–58.
11. For more information on A Three-Month Summer Rains Retreat Course, see (Nguyen 2007, pp. 62–63).
12. For more information on the modern school for monks See (Ninh 2020, pp. 198–217).
13. The list is compiled in the book (Như 2009).
14. For more information on the nun Dieu Khong see (DeVido 2014, pp. 71–82).
15. The poem in Vietnamese "Hành vi ngôn luận hẳn siêu quần/Vì cớ sao mà hiện nữ thân?/Trăm kiếp rèn nên gươm trí huệ/Một tay tháo sổ cũi phong trần/Dắt thêm đạo lữ năm ba lớp/Đánh vỡ tà sư sáu bảy phần . . . ."
16. Territorial unity was broken and traditional cultural values were lost due to the influence of Western civilization. The cooperative character of the Vietnamese villages was gradually eroded and a class of landless and land-poor Vietnamese grew. All armed resistance had been quelled and the infant mortality rate was consistently high. There are a few verses from an anonymous poem in Vietnam about the brutality of French colonialism: Rubber plantations is easy to go, difficult to return/ When you go, you are a strong man, when you come back, you are sallow and thin/ . . . . Rubber trees are green strangly/ Each tree fertilizes a worker's corpse. (Cao su đi dễ khó về/Khi đi trai tráng, khi về bủng beo . . . Cao su xanh tốt lạ đời/Mỗi cây bón một xác người công nhân). For more information on the negavite consequences See (Marr 1981). *Vietnamese Tradition on Trial, 1920–1945*: University of California Press: 15–53; See (Brocheux and Hémery 2001). *Indochine la colonisation ambiguë, 1858–1954*. Paris: Éditions la Découverte: 198–199.

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
