# Peer review of "The Rise of Vietnamese Nuns: Views from the Buddhist Revival Movement (1931–1945)"

_religions, doi:10.3390/rel13121189_

Round 1

Reviewer 1 Report

1/ Correct small mistakes  (line 23 : deveoped ; line 104 : womans duty ; line 122 : final point missing ; line 213 : « ngao chau » ?, line 253 : fromincurable) ; line 395 : double space)

 2/ Erase « in vietnamese » (line 25 and others)

 3/ Change paragraph : line 53 and give more explanations on this mendicant sect : Why this very small mendicant sect is considered as one main « branch » and not part of « theravada buddhism » ?

4/ Change paragraph line 55. The author should explain clearly that the renovation movement appeared in the colonial context. Then during the war different buddhists organizations appeared according to different states (Democratic Republic of Vietnam in 1945 ; the Associated State of Vietnam from 1949 ; then the Republic of Vietnam in 1955 and finally the Socialist Republic of Vietnam in 1976)

 5/ Line 72 : « began in the 1920s and ended in 1945 » :

I desagree with this argument, the renovation movement continued during the 1950 and even after. The author should make a clear distinction between 2 aspects : social aspect of female emancipation of nuns and political context (cold war) which changed the institutionnalization process (see comment 4).

 6/ Line 109 : add « Dai Nam » Thuc luc chinh bien

 7/ Foot note 26 : the author should make a clearer reference to the article quoted and add « our translation in english » in the text.

 8/ The author should make reference to Alec Soucy book (201) : The Buddha Side: Gender, Power, and Buddhist Practice in Vietnam

 9/ Line 273 : the author should explain and maybe balance the idea of « the decline of Buddhism ». What is reality a decline of buddhism (compare to which golden age ?) or a modernization of buddhism ?

 10/ Line 311 : For which reason did a new section was created in buddhist reviews in 1935 ?

 11/ Line 368 : « intellectual liberation » : would « intellectual emancipation » be more correct ?

 12/ It would be apprecited if the author could enlarge a bit the introduction to present the simultaneous evolution  of buddhist nuns in other buddhist countries (China, Japan, Siam for instance).

Author Response

Dear Sir/Dame,

First of all, I would like to thank you very much for your comments and for my article.

Combined with the second reviewer’s comments, I have edited part 1. Vietnamese women's situation during the French colonial period (1884-1945), part 2. The situation of the Vietnamese Buddhist nuns before the Buddhist revival movement. In part 2, I edited in the direction of clarifying Differences in nuns' situation/developments between the Three Regions of VN during colonial times on the basis of exploiting data from archival documents, Buddhist newspapers as well as the nun's memoirs. I have also rewritten almost all of part 3. Buddhist revival movement (attached file).

Religions Editorial gave me one month (October 17th - November 17th), but with major revision and I have just given birth, my article has not been refined in terms of writing style. I look forward to your understanding. While I wait for your comments, I will continue to improve my manuscript.

Sincerely,

Reviewer 2 Report

Please see the attached file and the comments that follow, they are two different sets of comments/suggestions.

I learned a lot from this article, it makes important discoveries for the field.

The article will be even better with the following revisions:

First, please see my comments in the PDF of the manuscript, Sticky notes and other comments.

Then, here are the most important issues to address:

* Engagement with sources as well as recent scholarship

Archives: It’s great that you used Archives nationales d’outre-mer. Yet I was surprised that you found only one survey? ANOM, RSTNF 2405. Please give more information about this collection/article/survey. Because in the abstract, I expected this would be a major source, but it turned out not the case. What happened, can you explain more what you were searching for, your archives experience, even in a footnote.

Depending on what your editor wants, find out the correct citation style for Archives.

Actually in this article author only referred to this source a few times, unless I’m mistaken. Can you make it clearer its importance to your work? Quote from it more.

Secondary: The Vietnamese secondary sources you use are recent and for the most part solid, though sometimes the Marxist tone (struggling; feudalism; dark and backward rural areas) of the recent Vietnamese sources colors the author’s narrative. The author could make it clearer what is the author’s historical point of view and what is the view/bias of the sources (no matter what language or author, VN, English, French…) Historians need to distinguish our voice (our analysis) from the sources and this author does it well when she quotes from the nuns’ writings.

Buddhist journals are also one of your main sources,

Please note what issues of Từ Bi Âm, Viên Âm and Đuốc Tuệ did you look at, like: Volume what to what? (Years...)

But there were many other such journals in the colonial era, did you look at them?

It's great to translate and quote the nuns’ poetry! This is excellent. Why were nuns writing poetry, tell the reader. We know that poetry is such an important form in VN literature and in VN Buddhism as well but why? Ven. Diệu Không wrote lots of poetry, etc etc. The more we hear nuns’ voices, the better. And notice how the poetry was talking about they hoped for more opportunities, equal opportunities, to become trained, to be good Buddhist practitioner/monastic, etc, instead of trying to cram your excellent evidence into the modernist discursive box of struggle and oppression. Can you add more nuns’ voices?

Also there are so many VN language sources about the chấn hưng phật giáo việt

 See the following sources, they refer to many other relevant works you can also consider.

This article needs more engagement with English-language works! A brief reference to Shawn McHale’s work is not enough. You will find sources in these articles including reference to Vietnamese sources that might be helpful. Ven. Diệu Không belonged to the Vietnamese royal family, and the roles of royal Buddhist women are very important for your story, please see the chapter “Eminent Nuns in Huế, Vietnam.” Eminent Buddhist Women, edited by Karma Lekshe Tsomo. Albany, NY: State University of New York Press, 2014, pp. 71-81.

In this book there two other chapters on nuns in Vietnam.

Please see this book Modernity and Re-enchantment in Post-Revolutionary Vietnam, edited by Philip Taylor. The information and sources in this book and this chapter might strengthen your article: “Buddhism for this World: The Buddhist Revival in Vietnam, 1920-51 and its Legacy.” Modernity and Re-enchantment in Post-Revolutionary Vietnam, edited by Philip Taylor. Singapore: Institute of Southeast Asian Studies 2007 and Lantham, MD: Lexington Books 2008, pp. 250-296.

Differences in nuns’ situation/developments between the Three Regions of VN during colonial times: This is really important and please make it even clearer to the reader

Organization: Pretty clear but…
1. Introduction

1.     Why another “1?” Vietnamese Women’s situation…

However you mention Revival Movement earlier than Part 3, the section on BRM, so whenever you first mention a person or movement etc give a brief explanation then tell reader something like “to be discussed in more detail in section of this paper.”

2.  The situation of VN nuns before the Buddhist Revival Movement

3. The Buddhist Revival Movement

4. Conclusion

Author Response

Dear Sir/Dame,

First of all, I would like to thank you very much for your material and content suggestions for my article. Your comments help me see and evaluate the research problem more clearly and objectively.

I have edited part 1. Vietnamese women's situation during the French colonial period (1884-1945), part 2. The situation of the Vietnamese Buddhist nuns before the Buddhist revival movement. In part 2, I edited in the direction of clarifying Differences in nuns' situation/developments between the Three Regions of VN during colonial times on the basis of exploiting data from archival documents, Buddhist newspapers as well as the nun's memoirs. I have also rewritten almost all of part 3. Buddhist revival movement (attached file)

Religions Editorial gave me one month (October 17th - November 17th), but with major revision and I have just given birth, my article has not been refined in terms of writing style. I look forward to your understanding. While I wait for your comments, I will continue to improve my manuscript.

Sincerely,

Round 2

Reviewer 2 Report

The re-organization of the article is improved greatly, and the author did address my earlier concerns about why define the Buddhist Revival 1931-1945; to engage with more English sources; to talk about the Royal women and Buddhism; and to keep an authorial distance between polemical secondary scholarship; and compare the three regions, very interesting.  Also the discussion of her sources is very helpful. 
I recommend some more revisions as follows:

Line 19, please define Nun, in the VN context

Line 20, "number of nuns is greater than monks": when, what years? any more information about it?

when you say funds do you mean archives (Fonds?)
Note 5: It's good you added more Western scholarship but could you quote from them a bit more later, either about the Colonial period or the Revival?

Line 43, just to say "1930s" is ok

LIne 80: why cant documents be retrieved?

Line 81, Line 160, what are Notices? 

Line 82, took advantage of, do you mean utilize

LIne 85, Feudal, please define it, this is Marxist historiography. Which is fine but please define what it means in terms of Vietnam.

LIne 115, is gaining popularity, do you really mean the present tense?

Line 158, their temple or temples?

Are these lines something you wrote or are you quoting? Can you explain more clearly?

However, at the Buddhist level, the vast majority were at the elementary level, and even many nuns were in the situation of being “illiterate” and “knowing the prayers by heart”. 

Lines 203-4 The Buddhadharma is the Buddhadharma itself? (not clear)

Line 210 limited to the village, what does that mean?

Line 219 female officials are who? Like women who work in the Royal Household? 

LIne 221 "into the pagoda,"    what pagoda?

Line 221, you can just say Annam not "the Annam"

Note 40: it's tragic indeed about Huê Tâm

Lines 265-276 this is great, yet why indented and please revise the English a bit 

Line 265: not "real name" but "secular name" 

Note 47 I dont understand it

Lines 398-402  Please comment more on these lines. Here are supposedly reformist nuns speaking about women's roles in a conservative Confucian way! can you discuss why nuns would use this type of argument, albeit with the modern idea of nationalism and nation building. And in fact Chinese Buddhist reformers argued in a similar fashion.

Lines 389 and 503, what are evil monks?

Note 93, please add more, how about refer to one of David Marr's books?French colonialism was brutal at times.

Conclusion should not be numbered 4, because 4 is Buddhist Revival section.

Section 4, Buddhist revival: please add discussion of the VN Buddhist Revival in the context of Buddhist revivals taking place in Asia, for example see The Influence of Chinese Master Taixu on Buddhism in Vietnam (Elise DeVido), Journal of Global Buddhism, 2009. (can download from internet)

Other: Nguyen Lang is Thich Nhat Hanh, as I guess you know.

Can you please re-work the title and abstract, they seem a bit rushed.

*Please read the chapter on Eminent Nuns in Hue, here enclosed. It doesnt seems like the author had it in hand.

Author Response

Dear Sir/Dame,

I have added my article the points according to your suggestions.

English writing style was corrected by Religions Journal’s English editor.

The revisions are highlighted by Track change.

Sincerely,
